# The Use of Biological Sensors and Instrumental Analysis to Discriminate COVID-19 Odor Signatures

**DOI:** 10.3390/bios12111003

**Published:** 2022-11-11

**Authors:** Vidia A. Gokool, Janet Crespo-Cajigas, Amritha Mallikarjun, Amanda Collins, Sarah A. Kane, Victoria Plymouth, Elizabeth Nguyen, Benjamin S. Abella, Howard K. Holness, Kenneth G. Furton, Alan T. Charlie Johnson, Cynthia M. Otto

**Affiliations:** 1Global Forensic and Justice Center, Department of Chemistry and Biochemistry, Florida International University, Miami, FL 33199, USA; 2Penn Vet Working Dog Center, Clinical Sciences and Advanced Medicine, School of Veterinary Medicine, University of Pennsylvania, Philadelphia, PA 19104, USA; 3Department of Emergency Medicine and Penn Acute Research Collaboration, University of Pennsylvania, Philadelphia, PA 19104, USA; 4Department of Physics and Astronomy, University of Pennsylvania, Philadelphia, PA 19104, USA

**Keywords:** canine detection, COVID-19, VOCs, SPME-GC-MS, odor signatures

## Abstract

The spread of SARS-CoV-2, which causes the disease COVID-19, is difficult to control as some positive individuals, capable of transmitting the disease, can be asymptomatic. Thus, it remains critical to generate noninvasive, inexpensive COVID-19 screening systems. Two such methods include detection canines and analytical instrumentation, both of which detect volatile organic compounds associated with SARS-CoV-2. In this study, the performance of trained detection dogs is compared to a noninvasive headspace-solid phase microextraction-gas chromatography-mass spectrometry (HS-SPME-GC-MS) approach to identifying COVID-19 positive individuals. Five dogs were trained to detect the odor signature associated with COVID-19. They varied in performance, with the two highest-performing dogs averaging 88% sensitivity and 95% specificity over five double-blind tests. The three lowest-performing dogs averaged 46% sensitivity and 87% specificity. The optimized linear discriminant analysis (LDA) model, developed using HS-SPME-GC-MS, displayed a 100% true positive rate and a 100% true negative rate using leave-one-out cross-validation. However, the non-optimized LDA model displayed difficulty in categorizing animal hair-contaminated samples, while animal hair did not impact the dogs’ performance. In conclusion, the HS-SPME-GC-MS approach for noninvasive COVID-19 detection more accurately discriminated between COVID-19 positive and COVID-19 negative samples; however, dogs performed better than the computational model when non-ideal samples were presented.

## 1. Introduction

The outbreak of severe acute respiratory syndrome coronavirus 2 (SARS-CoV-2) began affecting people in the United States (US) in the winter of 2019. Infection with the virus has resulted in the deaths of over 1 million individuals in the US and over 6 million globally [1]. It has been determined that the virus primarily travels via airborne respiratory droplets, typically generated during talking, coughing, and sneezing [2,3]. SARS-CoV-2 is generally spread when aerosolized droplets from an infected person come into contact with a healthy person’s mucous membranes (i.e., the eyes, nose, or mouth) or through direct physical contact between an infected and uninfected person [4]. Some infected patients may present as asymptomatic [5], while other cases involve fever, cough, and unexplained fatigue [6,7]; more severe cases can result in hospitalization for respiratory failure [7], heart failure [8], or sepsis [9]. Since SARS-CoV-2 is capable of rapid transmission, and has proven to result in drastic health consequences, there is a pressing need for effective surveillance testing.

Currently, the most sensitive testing methods for SARS-CoV-2 are fairly time consuming, and the rapid tests can fail to detect low viral loads. Nucleic amplification detection methods such as reverse-transcriptase polymerase chain reaction (RT-PCR) are the gold standard for the diagnosis of SARS-CoV-2. Although RT-PCR testing has a relatively high sensitivity, false negatives can occur for several reasons, including viral content below the test’s limit of detection, poor sample collection, and faulty technique in sample testing [10]. False negative results can lead to increased spreading of the virus from unknowingly infected individuals [11]. Additionally, those with asymptomatic COVID-19 expression may not proactively seek out testing due to their lack of symptoms [12].

Due to disease state misdiagnosis and lack of detection in asymptomatic carriers, there is a persisting need for the deployment of noninvasive population screening for COVID-19. This study investigates two prospective screening methodologies applied to clothing worn by individuals in the community who had been tested for COVID-19: the first is the use of trained detection dogs, and the second is the analysis of samples via headspace-solid phase-microextraction-gas chromatography-mass spectrometry (HS-SPME-GC-MS). These noninvasive biological and instrumental systems were applied in parallel to discriminate the complex odor signature from COVID-19 body odor samples.

Current research demonstrates that dogs can be trained to detect hypoglycemia [13], lung cancer [14], ovarian cancer [15,16], prostate cancer [17], and urinary tract infections [18]. More recently, several research groups have trained dogs to identify signature volatile organic compounds (VOCs) of SARS-CoV-2 in a variety of mediums; including pharyngeal secretions [19], urine [20], saliva [20], sweat [21,22], and breath [23,24]. However, the distinct VOCs associated with the SARS-CoV-2 virus have been minimally explored outside of VOCs in exhaled breath [25,26]. VOCs are produced by cells during metabolic processes, inflammatory processes, and oxidative stress. These compounds are released from the human body via breath, urine, feces, and blood [27]. While several studies have identified key COVID-19-associated VOCs in breath [23,24,25,26,28,29,30], it can be risky to utilize breath as a medium for detection in dogs, since breath from positive patients contains live virus [28,31] and dogs are susceptible to contracting COVID-19. In contrast, sweat from COVID-19 positive patients is a safer sample source, containing no live virus [32].

VOCs associated with the SARS-CoV-2 virus in sweat samples can be identified using headspace analysis via solid phase microextraction (SPME), which allows for the capture of gaseous analytes onto a coated sorbent surface, the SPME fiber [33]. Once on the SPME fiber, the analytes can be transferred to another instrument for desorption and analysis. HS-SPME is often used in conjunction with gas chromatography-mass spectrometry (GC-MS), where the GC-MS separates and analyzes the compounds captured by the HS-SPME procedure. Various works have been published on the use of HS-SPME-GC-MS for the detection and characterization of VOCs from human specimens, including sweat, blood, saliva, urine, breath, hair, and fingernails (see [34] for a review). Complementary detection of VOCs by biosensors (dogs) and analytical instrumentation (HS-SPME-GC-MS) can narrow down areas of volatile biomarkers [16] and allow for further characterization of these unique VOC profiles, providing a critical step forward for the detection of COVID-19.

In this study, we compared the performance of dogs and the headspace analysis of COVID-19 samples by HS-SPME-GC-MS. Both methods used T-shirts containing body odor/sweat from both infected and healthy individuals, regardless of identifying factors such as gender, race, age, and geography. This work demonstrates the ability to create a multi-modal approach to complex odor detection and propels this conversation through the task of detecting COVID-19 presence.

## 2. Materials and Methods

### 2.1. Collection of COVID-19 Samples

#### 2.1.1. Approvals

All animal studies were reviewed and approved by the institutional animal care and use committee at the University of Pennsylvania (IACUC protocols 806922, 804900). Handlers of privately owned dogs that performed the detection tasks provided informed consent. The human study collecting T-shirts was reviewed and approved by the University of Pennsylvania Institutional Review Board (IRB protocol 843534). Participants provided informed consent.

#### 2.1.2. Human Participant Recruitment and Follow-Up Health Survey

People recruited on social media (from July 2020 to April 2021) were asked to participate in the study by wearing a T-shirt overnight if they had been tested for COVID-19 within the previous 48 h. Participants filled out a survey to determine their eligibility. Data collected included: age, type of COVID-19 test performed, whether the test was performed within the preceding 48 h, COVID-19 test results (positive/negative/inconclusive) and when the results were received, presence of symptoms in the past 7 days, COVID-19 vaccination history, contact information, T-shirt size, how they heard about the study, and their preferred method of communication. Potential candidates were disqualified if they were younger than 18 years of age, not living in the United States, allergic to cotton, did not provide contact information, were tested via rapid antigen COVID-19 test instead of a PCR test, or if they were not tested for COVID-19 within the past 48 h.

Eligible participants were shipped an unwashed cotton T-shirt, assigned a case identification number (case I.D.), and were categorized as: positive symptomatic, positive asymptomatic, negative symptomatic, or negative asymptomatic. Informed consent was obtained via electronic signature (Qualtrics) or a scanned copy of the consent form with a written signature. Participants provided a de-identified copy of their PCR test results.

Following receipt of the T-shirt and instructions (wear it overnight without washing it or wearing any other type of nightshirt under it), participants were asked to complete a health survey confirming test results and symptomology, including duration that the shirt was worn, use of any fragrances, any humans or animals with whom they shared their bed, a rating of their current stress level on a scale of 1 (little) to 10 (maximal), current or past medical conditions, and current medications. The results of this health survey can be found in the supplementary information (Appendix A). The term “symptom count” is used to reference the number of separately accounted symptoms disclosed by the participant; for example, a participant who noted coughing and headaches as symptoms would have a symptom count of two.

#### 2.1.3. Preparation of Sample Collection Materials

All T-shirts were handled with nitrile gloves, individually numbered on the collar with a permanent black marker, and packaged in the same environment. The numbered T-shirt was sealed in a one-gallon Ziploc™ bag removing as much air as possible. The case I.D. number was on the Ziploc™ bag, as well as the enclosed instructions. The participant was instructed to write the date and time when the T-shirt was put on and when it was removed on the label affixed to the bag. The prepared collection materials were stored according to size, in boxes at room temperature.

Each participant was shipped (a) one labeled T-shirt of the designated size (in a Ziploc™ bag), (b) one labeled Tyvek return shipping bag, and (c) one set of instructions via overnight courier. The return shipment was made via a scheduled pickup at least 24 h after the shirt was removed, placed back in the Ziploc^TM^ bag, and sealed in the shipping bag to ensure a lack of infectivity [35].

T-shirts were shipped from participants back to the University of Pennsylvania between July 2020 and June 2021. The date the samples were sent to the Penn Vet Working Dog Center (PVWDC) was accounted for in the randomization when creating training and test sample sets, adding another layer of variability for the dogs to generalize across. Sample processing was performed by PVWDC and the Penn Acute Research Collaboration coordinator team at the University of Pennsylvania. All negative shirts were processed prior to any positive shirts. Fresh nitrile gloves were worn for handling each individual shirt. All surfaces were cleaned with isopropyl alcohol between shirts.

Shirts were cut into six pieces: (1–2) the front cut in half, (3–4) the back cut in half, (5) the right sleeve, and (6) the left sleeve. Once cut, the right side and left side pieces were separated, and each half of the shirt was placed into a mason jar labeled with the case I.D. number and the donor’s COVID-19 status (+/−). The right halves of the shirts were sent to Florida International University (FIU) and the University of Pennsylvania Johnson Lab to be utilized for VOC analysis or other COVID-19 studies, respectively. The left halves of the shirts were kept for canine use.

#### 2.1.4. Collected COVID-19 Positive and COVID-19 Negative T-Shirt Samples

Samples were collected from people across the US; only samples returned with a completed donor health survey were considered for canine training/testing. Two hundred and ninety-three (293) samples were collected, entered into the total dataset, and used for training purposes in the study. Additional samples were collected, but these were not used in the training/testing schemes. The majority of samples were obtained from participants who identified as White or Caucasian, which was accounted for when distributing samples throughout pre-determined training sets for canine training. The demographic distribution of the collected T-shirt sample inventory can be found in Table A1.

The sample portions allocated for canine use were sealed in appropriately labeled mason jars and stored in lockers kept at room temperature. Samples remained in the lockers before and after being utilized for canine training/testing. Additionally, samples were separated based on COVID-19 test results, so negative and positive samples were never stored adjacent to one another. In total, for the dogs, 220 samples were used in the training phases, and 73 samples were used in the test phase; donor demographics are shown in Table 1. Two COVID negative samples, #291 and #432, were included in both the training and test sample sets. All other samples were seen once by each canine.

T-shirt portions allocated for instrumental analysis using HS-SPME-GC-MS were sent to Florida International University (FIU) in labeled Ziploc^TM^ bags. In accordance with the safety protocols in place at the time of analysis, all T-shirt materials received by FIU were irradiated using a 10-min exposure to 254 nm ultraviolet (UV) light. This procedure was conducted to ensure that no live virus was present on the samples before they were handled.

### 2.2. Canine Odor Discrimination Methods

#### 2.2.1. Canine Participants

Five dogs (4 privately owned, one owned by the PVWDC) were selected to participate in the study (see Table 2). To participate in the study, dogs must have demonstrated (a) a “trained final response” (stand-stare alert for greater than 2 s, or a sit alert) to training odor presented on the wheel (see Figure 1), (b) the ability to sequentially search the scent wheel ports 1 through 8, and (c) a “blank wheel response” by leaving the wheel room or sitting on a platform when the target odor was not present in the wheel and all ports have been searched. All dogs were initially trained on wheel searching mechanics using the universal detector calibrant training odor (UDC) [36].

#### 2.2.2. Use of T-Shirt Samples in Training and Testing

Each week, after the dogs had been initially trained on the odor, the dogs were exposed to two training sets containing four negative and two positive samples, all unique and only used once throughout training. Both negative and positive samples contained only the sleeves of the collected T-shirts. The distribution of samples was randomized to ensure dogs were exposed to a range of ages, genders, ethnicities, races, and pre-existing medical conditions in each training session. A similar method of randomization was used to create the sets on which the dogs were tested at the end of their training. The majority of test sets contained 12 novel negative samples and 2–3 novel positive samples (Appendix A). Each of the selected samples varied in demographic categorizations and length of time T-shirt was stored between and within tests. Two previously seen negative samples (#291 and #432) were included in the test set, leading to a total of 59 negative samples across tests. A range of sample demographics was used to examine the generalization of the COVID-19 odor cue from person to person.

#### 2.2.3. Distractors

In addition to the positive and negative T-shirt samples, dogs were also presented with distractor items. These items were selected to ensure that the dogs were alerting on COVID-19 positive T-shirts and not on any untargeted scents/odors that potentially coincided with these samples. Items included an unworn T-shirt, a shipping envelope from the PVWDC, a shipping envelope sourced from the research assistant sending out shirts, latex gloves, nitrile gloves, isopropyl alcohol, and a permanent marker.

#### 2.2.4. Training Procedure

The training progression that was implemented while training each canine (Table 2) is outlined in Table 3 and described in further detail below.

#### 2.2.5. Odor Learning Phase

In the initial phase of the study, dogs were rewarded for sniffing the COVID-19 positive sample. Dogs were presented with jars containing the sleeves of shirts worn by COVID-19 positive patients as a target odor. When the dog sniffed this target odor, the behavior was marked with a conditioned secondary reinforcer (i.e., noise generated by a “clicker”) and paired with a food or toy reward. The target jars were covered with a mesh lid, allowing full odor release but preventing access to the material. In the same session, dogs were presented with jars containing the sleeves of shirts worn by people who were COVID-19 negative. Dogs were not rewarded, nor was their behavior marked for sniffing the negative sample.

During this phase, the COVID-19 negative jars were covered by a lid with 1 hole in the top, as compared to the fully open COVID-19 positive jars. This approach allowed the dogs to utilize two differences in odor for differentiation: first, the scent profile difference between COVID-19 negative and COVID-19 positive, and second, a concentration difference where the COVID-19 positive odor, covered with a mesh lid, was stronger than the COVID-19 negative odor, covered with a single-hole punch lid. The concentration difference functioned as “training wheels”, which were phased out as the dogs’ training progressed. This method is known as “errorless learning” [37] and has been used in a previous COVID-19 detection study from the PVWDC [20].

At the beginning of each training session during the odor learning phase, the dogs were presented with the COVID-19 positive and COVID-19 negative jars on the floor. The location was randomized—either left or right—across ten trials. The trainer clicked when the dog sniffed or showed a trained final response on the COVID-19 positive jar and was ignored when the dog sniffed or showed a trained final response on the COVID-19 negative jar.

#### 2.2.6. Odor Learning Sessions on the Wheel

After 10 trials with only two samples (1—positive, 1—negative) were completed, the COVID-19 positive and COVID-19 negative jars were moved onto the scent wheel. The scent wheel contains 8 total ports: two of these were the COVID-19 positive and COVID-19 negative odors, respectively, and the remaining 6 ports were distractor items (Figure 1). All trials were video recorded.

To prevent human cues and bias, the dog and trainer were blocked from the trial area to prevent visual observation of sample placement. The trainer remained behind a barrier during the trial, such that they were out of sight from the dog. During each trial, the dog was sent to search the scent wheel. A positive response to COVID-19 positive odor was marked by the trainer with a conditioned secondary reinforcer (e.g., a clicker), then rewarded with praise and food or a toy. False negatives (passing the target odor) and false positives (alerting at the incorrect odor) were ignored. If the dog passed the positive odor without presenting a trained response (false negative), they were sent back to the wheel without a reward to execute a new search for the positive odor.

For any given session in this phase, there were 4 to 6 “blank wheels” where there were control and distractor odors but no target odor present on the scent wheel. During a blank trial, dogs are to sniff and pass each of the eight ports and then exit the wheel room and sit.

When the dog achieved 90% accuracy (9/10 correct trials) across all scent wheel trials in one session, the dogs moved to the subsequent phase of training.

#### 2.2.7. Training Phase 1

In Training Phase 1, there was no longer exposure to a positive sample prior to the dogs’ wheel trials. Additionally, dogs were presented with double the number of positive and negative samples in each session. Per trial, the scent wheel contained one or zero COVID-19 positive samples and up to 4 COVID-19 negative samples. The other ports contained distractor items, which were covered by a mesh lid. The jars containing the COVID-19 positive shirts were covered with a mesh lid. The jars containing COVID-19 negative shirts continued to have one hole in the lid, again to restrict odor output.

Training sessions included multiple presentations of both targets and controls; however, only the initial response was used to calculate the required performance to move to the next phase.

For a dog to move to the next phase, it needed to have a minimum true positive response of 5/6 to positive targets on first presentation (83% sensitivity) and a minimum true negative response of 10/12 non-targeted odors on first presentation (83% specificity). Due to the number of samples, they had to complete a minimum of 3 sessions in any given phase to achieve this benchmark. This benchmark was applied to Training Phases 1–4.

#### 2.2.8. Training Phase 2–4

Phases 2–4 were procedurally identical to Phase 1. In Phase 2, the jars containing the COVID-19 negative shirts used lids that had four holes punched in the top. In Phase 3, the jars containing the COVID-19 negative shirts used lids that had 16 holes punched in the top. In Phase 4, the jars containing the COVID-19 negative shirts had mesh lids. The increase in holes in the COVID-19 negative jar lids slowly removed the concentration cue that was initially provided to help the dogs distinguish it from the COVID-19 positive odor. Training sessions included multiple presentations of both targets and controls; however, only the initial response was used to calculate the required performance to move to the next phase.

#### 2.2.9. Test Phase

Dogs completed five test sessions consisting of three trials each. The sessions were run double-blind (i.e., the handler did not know the location of the positive sample, and the experimenter left the room prior to the start of each trial); such that the dog would not inadvertently be cued to alert on any sample. Two of the test sessions contained one blank trial, in which there was no positive sample. The remaining sessions contained 3 trials with a novel COVID-19 positive sample each trial. As such, there were 13 total COVID-19 positive samples shown to the dogs, as well as 59 COVID-19 negative samples (Appendix A). The positive samples were purposefully selected to be diverse in race, age, vaccination status, time the shirt was collected, and the number of symptoms (See Table A2).

#### 2.2.10. Behavioral Analysis

The test phase data was behaviorally coded (translated into numerical values) by two research assistants using BORIS [38]. The coders recorded the duration of time each dog spent at each odor port for each trial in the 5 test sessions. The coders were unaware of the location of the positive sample.

#### 2.2.11. Statistical Analysis

A binomial mixed-effects logistic regression with sample type as the fixed effect and dog as a random effect was conducted to determine the effects of a COVID-19 positive or COVID-19 negative status of a sample on the duration of time spent at port for each dog. Analyses were carried out in R using GLMER [39].

### 2.3. HS-SPME-GC-MS Methods

#### 2.3.1. Sample Set Demographics

A subset of the samples used in the dog trials were transferred to FIU for HS-SPME-GC-MS analysis. Due to this subgrouping, the HS-SPME-GC-MS samples reflect a different distribution of demographics than are reflected by the dog trial samples (Table 1). Table 4 shows a demographic breakdown of the donors contributing to HS-SPME-GC-MS characterized T-shirt samples. Eleven (11) sample donors did not provide demographic information.

#### 2.3.2. HS-SPME-GC-MS Analysis Procedure

Following sample irradiation, T-shirt samples were prepared for reagent-free sampling by removing a 4 × 4 inch piece of fabric from the underarm section using sterilized scissors. This piece was then placed inside a 40 mL headspace vial, which was pre-cleaned according to the methods published by Gokool et al. (2022) [40]. Samples were stored at room temperature (20 °C) between sample collection and analysis.

Samples were placed in a digital heating bath set at 50 °C immediately prior to sample extraction. A clean 50/30 µm divinylbenzene/carboxen/polydimethylsiloxane (DVB/CAR/PDMS) SPME fiber was exposed to the headspace of the T-shirt samples at a 1-inch fiber exposure setting. No previous sample equilibration was performed. After 24 h, SPME fibers were unexposed and removed from the sample headspace.

Analytes on the SPME fibers were thermally desorbed at 270 °C for 4 min (3-inch fiber height) into the heated inlet of the GC (Agilent 8890; Agilent Technologies, Santa Clara, CA, USA). A splitless injection method with a 1 mL/min column flow was implemented on a HP5-MS UI capillary column (15 m × 0.250 mm I.D. × 0.25 µm phase thickness; Agilent Technologies). Helium was used as the carrier gas. Oven temperature parameters started at 40 °C (1.25 min hold), increased to 165 °C (5 °C/min rate), and concluded at 270 °C (30 °C/min rate). The total method runtime was 29.75 min. A mass spectrometer (Agilent 5977B MSD; Agilent Technologies) with an electron impact ionization (EI) source and quadrupole mass analyzer was used with the following parameters: MS source was maintained at 230 °C, MS Quad at 150 °C, transfer line at 280 °C, EI source at 70 eV, and scan range at m/z 50–550.

#### 2.3.3. Statistical Analysis

A semi-quantitative approach was taken for the analytical procedure because of the novelty of the samples collected (bodily emanations from individuals who tested as positive or negative for COVID-19). Due to a lack of pre-existing knowledge of the sample and sample matrix, the researchers opted for a pattern recognition-based approach to data analysis.

Linear discriminant analysis (LDA) was used to model the class separation of COVID-19 positive/negative samples. LDA is a dimensionality reduction technique that is used for supervised classification problems. The technique creates a linear combination of the submitted features to form separation between the classes.

#### 2.3.4. Data Pre-Processing

An untargeted approach was applied to the dataset to uncover underlying relationships and their contributing compounds. All 148 received T-shirt samples were analyzed using HS-SPME-GC-MS. The resulting GC-MS chromatograms were pre-processed using a proprietary piece of software developed at FIU; this software has previously been applied in the pre-processing of human odor data collected by HS-SPME-GC-MS [40]. The collective dataset was evaluated for compounds eluting at approximately the same time. In addition to the 148 COVID-19 positive/COVID-19 negative samples, three blank unworn T-shirts were sampled to establish background signals from the fabric substrate. Compounds that were present in all the background samples were removed. The following calculations were determined using a dataset consisting of square-root transformed, total ion chromatogram (TIC) peak areas for retention time-aligned peaks.

## 3. Results

### 3.1. Canine Behavioral Coding Interrater Analysis

An inter-rater analysis was run in R to determine the reliability of the participating video coders. Six randomly chosen videos out of the 25 total test videos (24%) were used for this analysis The inter-rater analysis was run based on a single-rating, consistency, two-way mixed effects model. The model displayed an intraclass coefficient of 0.843, with a 95% confidence interval of 0.793 to 0.882. These findings indicate that the video coding was performed with good to excellent reliability [41].

### 3.2. Canine Training Data

Dogs spent different amounts of time in each training phase prior to achieving the benchmark to be tested (see Figure 2). In the last phase, where both the negative and positive samples had fully open lids, there was a divide between Roxie and Rico, who spent very little time (5 sessions) in this phase as opposed to earlier phases, and Tuukka, Toby, and Griz, who spent the most time in this phase (12, 13, and 11 sessions, respectively).

### 3.3. Testing Data

This study examined dogs’ ability to detect COVID-19 positive odor samples on T-shirts. An analysis was performed to assess whether dogs alerted significantly more often on the COVID-19 positive samples than the COVID-19 negative samples. An alert was characterized as a two-second stand-and-stare alert (Griz, Toby, Tuukka, and Rico) or a sit at the positive sample (Roxie). Dogs are more likely to alert on the positive samples than the negative samples, X^2^ = 2.92, z value = 8.645, *p* < 0.0001. Table 5 shows the sensitivity and specificity of each dog in the study.

#### 3.3.1. Factors Affecting Canine Behavior and Alert on Positive Samples

The authors examined how the dogs’ alert duration on positive samples was affected by demographic and symptomatic factors reported by the T-shirt sample donors. A linear mixed-effects regression with gender, age group, number of symptoms, and pet-owning status as the fixed effects and dog as a random effect was performed. None of these factors significantly affected alert duration (gender: F(56) = 1.11, t = 1.62, *p* = 0.11; age group: F(56) = −0.24, t = −1.06, *p* = 0.29; symptom count: F(56) = −0.01, t = −0.06, *p* = 0.95; pet ownership: F(56) = −0.64, t = −1.06, *p* = 0.29). Table 6 shows the results for alerts to positive test samples organized by the trial run. Table A2 depicts the dogs’ alert rate to the positive test samples; this information is organized by sample feature/characteristic (i.e., gender, age, etc.).

#### 3.3.2. Factors Affecting Canine Behavior and Alert on Negative Samples

Additionally, the authors examined how the dogs’ alert duration at negative samples was affected by demographic factors reported by the study participants. A linear mixed-effects regression with gender, age group, and pet-owning status as the fixed effects and dog as a random effect was performed. None of these factors significantly affected alert duration (gender: F(279) = −0.04, t = −0.19, *p* = 0.84; age group: F(279) = −0.05, t = −1.28, *p* = 0.20; pet ownership: F(279) = 0.06, t = 0.49, *p* = 0.62).

#### 3.3.3. Factors Affecting Canine Behavior and Alert on Positive or Negative Samples

A linear mixed effects model of duration at port with sample type (positive or negative) and sample number as fixed factors was created. The sample number denotes the order in which the samples were received over time. There is a main effect of sample type, B1 = −1.11, t(170) = −3.64, *p*= 0.0004, meaning canines spend statistically different amounts of time at the port for a positive sample (holding in place to alert) versus a negative sample (short interaction, no alert).

Importantly, there is no interaction between sample type and sample number, B1 = −0.001, t(170) = −1.03, *p* = 0.3. The earlier positive and negative samples did not have significantly different durations at odor than the later positive and negative samples. The time at which a sample was collected did not significantly affect the time spent alerting (for a positive) or passively sniffing (for a negative sample).

### 3.4. HS-SPME-GC-MS Results

Initial analysis of the formed dataset revealed a separation of data points. The observed separation is believed to be driven by one or more unidentified class characteristics. As seen in Figure 3, there is subgrouping occurring within the dataset that has not been attributed to a sole influence or characteristic, such as COVID-19 status. The separation may be due to a confounding effect or a combination of class characteristics. The characteristics which have been investigated and ruled out as sole contributors to the dataset grouping are gender, age, vaccination status, and pet ownership. An insight into subgrouping composition was achieved when viewing the dataset in terms of self-identified race.

The consideration of donor race indicated that all donors in one subgroup (right) share the racial identification of “White or Caucasian.” Due to this observation, the following co-interactions were considered for further explanation of data subgrouping: (a) race and COVID-19 status, (b) race and symptomatic presentation, (c) race and gender, (d) race and age, and (e) race and vaccination status. Although these co-interactions were investigated, they were not seen to provide any further explanation for the observed phenomenon.

The peaks of interest contributing to the subgrouping (i.e., the split observed) seen in Figure 3 were identified (by retention time) and removed from the dataset. This action constituted a removal of 15 peaks of interest. This caused the dataset to decrease from 126 features of interest to 111 features of interest. The removal of this subsetting factor mitigates the risk of building an instrumental model off erroneous information, which is not related to the COVID-19 positive/negative status of a donor. The refined dataset no longer demonstrated the previously observed, unexplained subgrouping indicating that the developed instrumental model would not be informed by the unidentified effect (Figure 4).

#### Model Development

Following the pre-processing procedure described above, the dataset containing retention-time-matched peaks of interest was square-root transformed. Note that the dataset was refined to remove peaks found to be present in all of the analyzed background samples (unworn T-shirts) and peaks that contributed to the subgrouping viewed in Figure 3. This dataset was used to build a linear discriminant analysis (LDA) model using JMP^®^, Version 16.1.0. (SAS Institute Inc., Cary, NC, USA 1989–2021). LDA is a supervised learning technique that uses a linear combination of features of interest (variables) to discriminate between classes. LDA allows for linear modeling of the COVID-19 positive and COVID-19 negative samples using the TIC peaks of interest (identified in the GC-MS data) to separate and differentiate between COVID-19 status in the sample set.

The LDA model displayed in Figure 5 demonstrates the separation of 54 COVID-19 positive and 94 COVID-19 negative donor profiles. The displayed model was seen to perform moderately when all data points were used to train it; true positive rate (TPR) = 96.3%, true negative rate (TNR) = 96.8%, false positive rate (FPR) = 3.2%, and false negative rate (FNR) = 3.7%. This model resulted in five misclassified samples with three False Positives and two False Negatives. When leave-one-out cross-validation (LOOCV) was applied to the model, the performance was very poor, with TPR = 57.4%, TNR = 61.7, FPR = 38.3%, and FNR = 42.6%. LOOCV was performed using R-3.6.1, an open-source software.

Further review of the misclassified donor samples, seen in the unvalidated model, revealed that the five misclassified individuals retained the class characteristic of “sharing their bed with at least one pet cat or pet dog”. The researchers believe that the transfer of pet fur and dander onto the collection substrate influenced the collected volatile organic compound profiles. This pet fur and dander was visible on several of the samples collected with this donor characteristic, as can be seen in Figure A1.

It was beyond the scope of this project to train a predictive model to differentiate between COVID-19 positive and COVID-19 negative samples despite the presence of animal interferents. As such, individuals who attested to sharing their bed with their pet animal(s) were removed from the dataset informing the LDA model. With the removal of these individuals, the model was tuned and cross-validated, demonstrating considerably less ambiguity than the previous iteration. The exclusion of individuals who shared the bed with their pet(s) reduced our sample population to 91. The updated model included 80 features of interest, reduced from the 111 features used to develop the model in Figure 5. There is a clear delineation between COVID-19 positive and COVID-19 negative donor profiles, as expressed in Figure 6.

The LDA model displayed in Figure 6 demonstrates the separation of (33) COVID-19 positive and (58) COVID-19 negative donor profiles. Both the non-validated model (informed by all data points) and the LOOCV model performed with a TPR = 100%, TNR = 100%, FPR = 0.0%, and FNR = 0.0%. The model did not “miss” any COVID-19 positive individuals. The donors in this subset stated that they did *not* share a bed with their pets. However, these donors did not indicate ownership of a pet. The previously discussed splitting of the dataset (seen in Figure 3) was not observed to be linked to pet ownership or sharing a bed with a pet.

## 4. Discussion

This study demonstrates a multi-modal comparative approach to the detection of COVID-19; the researchers utilized both dogs and HS-SPME-GC-MS to discriminate between COVID-19 statuses using human sweat samples. One of the challenges of many prior COVID-19 detection studies was that dogs were trained with relatively small sample sets, which lacked substantial demographic variation and focused on hospitalized COVID-19 patients [20,42,43]. Hospital populations are likely to be more severely affected by COVID-19 and be isolated during care. Whereas asymptomatic or mildly affected individuals are more likely to remain in the community and contribute to the spread of the virus. As such, their inclusion in the sample population is integral to identifying COVID-19 positive individuals in the populace. Another constraint posed by small sample sets, but evaded in this work, is the limited ability to determine the dogs’ capacity in generalizing to a broader donor population regarding age, race, and severity of illness, among other factors.

Both the dogs and the HS-SPME-GC-MS approach were able to discriminate COVID-19 positive from COVID-19 negative in this community-based population. Obtaining samples from a home environment led to an unexpected confounding factor: the presence of animal hair on the T-shirts. It is important to note that in this study 38.5% of individuals reported sharing their beds with a pet. The dogs’ performance was not affected by the presence of animal hair; however, the HS-SPME-GC-MS-informed model was. Despite the presence of animal hair interferents, the results of the study were consistent with the finding of Grandjean et al. (2022), who similarly tested dogs on samples from 335 individuals from two community screening centers [44]. Recent publications by Chaber et al. (2022) and by Grandjean et al. (2022) tested canine discrimination between COVID-19 positive and COVID-19 negative using sweat samples collected from France, UAE, and Australia [43,44]. These studies recounted high canine sensitivity and specificity, even reporting dogs could identify positive cases before the PCR test. These studies suggest that dogs could prove to be an effective population screening method [14,17].

Although the dogs were able to correctly identify and alert to COVID-19 positive samples and distinguish these amongst COVID-19 negative samples in this study, there was considerable variability observed in the dogs’ detection accuracy. Overall, the dogs gave correct responses (alerting on positive samples and ignoring negative samples) to 62.5% of the presented positive samples and 90% of the presented negative samples. When the canines were regrouped into high-performance (2/5) and low-performance (3/5) canines, a clear delineation in performance was seen. The high-performance group produced correct responses (alert to positive/pass negative) to 85% of positive samples and 94.8% of negative samples. In contrast, the low-performance group had correct responses to only 46.2% of positive samples and 87.2% of negative samples.

It is not always clear which dogs will perform well on specific odor detection tasks, and canine performances can vary widely (see Gonder-Frederick et al. (2017) for an example in diabetic alert dogs [45]). Canine performance is known to vary based upon several factors, including canine-oriented factors such as personality traits [46,47], specific polymorphisms in olfactory receptor genes [48], and handler-oriented factors, including the identity and familiarity of the handler [49] and the handler’s belief about the odor’s location and positive/negative status [50]. Identification of the specific VOCs may help provide the tools to evaluate the effects of canine-oriented factors. Further research can explore how to identify potentially high-performing detection dogs prior to training, to prevent significant loss of time and capital.

Importantly, the dogs did not display a significantly different performance in detecting COVID-19 within a specific demographic group, nor was their performance affected by the storage time of the samples used. The dogs were trained on a large number of samples from a variety of demographic groups. Additionally, dogs saw samples that ranged from 298 days (oldest negative sample seen by last tested dog) and 283 days (oldest positive sample seen by last tested dog) to 21 days (newest negative sample seen by first tested dog) and 63 days (newest positive sample seen by first tested dog), in terms of when the odor/scent was collected on the shirt to the time they were presented to the dogs (See Table A3). The consistent performance of the dogs suggests that the sample collection and storage procedures (collect onto cotton T-shirts and store in a sealed container at room temperature) produce a sample with a stable odor profile and non-depleted headspace through long periods of static storage. The canines’ performance in testing reinforces the idea that training detection dogs using a diverse set of samples supports their ability to generalize odors (see [51] for a similar concept in machine learning).

HS-SPME-GC-MS was used as a reagent-free technique for the capture and detection of VOCs present in human body odor samples. The human scent profiles obtained through this procedure were used to develop an LDA model for the predictive classification of COVID-19 status, informed by sweat expression. In this experiment, the developed LDA model depicted in Figure 5 was seen to perform very poorly in comparison to the model shown in Figure 6. The difference in model performance was linked to the inclusion of samples collected from donors who reported sharing a bed with their pet(s). The removal of these individuals from the LDA model allowed for the creation of a predictive model that had no misclassifications when cross-validated using a leave-one-out approach. The necessity of the removal of these individuals is contrasted by the results seen in the dog scent work, where the dogs were able to differentiate COVID-19 positive and negative samples regardless of the presence of pet interferents. The new predictive model for the differentiation of COVID-19 positive and COVID-19 negative T-shirt samples was informed by a sample set of 91 participants who did not share their beds with a pet. The model contained 80 features of interest, indicating a high degree of complexity in the characteristic VOC profile representative of COVID-19 expression in sweat. At its best performance, the developed LDA model (Figure 6) had a TPR = 100%, TNR = 100%, FPR = 0.0%, and FNR = 0.0% (validated using LOOCV). The contributing sample set, however, did not retain the same degree of diversity as the original sample population (reduced from 148 to 91 samples). The model requires further training, including the submission of more variation in donor demographics, if it is to be implemented in a non-academic setting.

### 4.1. Contrasts between Instrumental and Biological Methods Used for COVID-19 Positive/Negative Odor Discrimination

The duality of the methodology, using dogs and conventional analytical instrumentation, creates a parallel view of the strengths and challenges posed by each approach. Both techniques were demonstrated to be capable of discriminating between COVID-19 status when informed by human odor signatures. The use of scent work dogs demonstrated a method that allowed for the training of a detection system (dog) that was able to discriminate between samples sourced from a range of genders, races, and age groups, among other factors, without a demonstrated bias in their decision-making. However, while the training method utilized in this study proved to be effective for this set of participants, it also revealed variations in odor recognition between dogs, such as behavioral responses, that could not be controlled for. In a canine approach, the dog is the sensor, and unlike analytical instruments, canine abilities are highly variable between animals.

In contrast, the analytical approach was HS-SPME-GC-MS, a technique that uses components that are all commercially available and is easily reproduced. The LDA model, which was informed by the HS-SPME-GC-MS data, demonstrated a “perfect” determination of COVID-19 status. While the performance was superior to that of the dogs, the approach exhibited drawbacks of its own. In the context of canines being used as sensors, the dogs were calibrated to interpret odors sourced from domesticated animals, more specifically canine fur and dander, as background and not of interest to their task of identifying COVID-19 positive T-shirt samples. Dogs have previously been noted to distinguish specific target odors in environments with abundant and varied distracting odors [52]. The disparity in performance between the LDA model and dog trials speaks to the fact that while the dogs were trained to ignore non-targeted odors in their training, the LDA model was still in its training phase and had yet to learn how to disregard animal interferents in the dataset.

Another notable difference between the techniques is the issue of time. Using HS-SPME-GC-MS for status determination requires considerably more time per sample than required when using trained detection dogs. Excluding method development and training periods, the final procedure for determining if a single sample was COVID-19 positive or negative would require approximately 25h using HS-SPME-GC-MS; meanwhile, all five trained canines could be run on a single scent wheel line-up containing the sample in 10 min.

### 4.2. Limitations

The sample sets used to train/test the canines and computational model both skewed toward the same demographic categories. In terms of race, the sample sets were predominantly white or Caucasian, and in terms of gender, they were predominantly female. Additionally, the overarching dataset of total collected T-shirt samples lacked contributions from non-binary, genderqueer, and non-cis-gendered individuals.

While conducting a crowdsourcing campaign for sample donations allowed for a large number of samples to be collected from a geographically diverse set of donors, the procedure required some sacrifice of oversight into how the samples were handled pre-processing. The T-shirt samples used in this study were handled by multiple individuals from sample kit preparation to its final analysis, either by canine or HS-SPME-GC-MS analysis. At each point of interaction, the opening of containment vessels creates an opportunity for VOCs of interest to escape and interferents to settle onto the samples. In addition, the samples were stored in permeable membranes (Ziploc™ bags) from the time of collection until either (a) they were transferred to a glass mason jar and kept for storage by PVWDC or (b) they were transferred to FIU. Samples that were allocated for FIU remained in the permeable containment at room temperature until arriving at the facility. The storage of samples at room temperature or in permeable containers allows for the continuous emission of highly and nominally volatile compounds from the sample. The long-term storage of samples under these conditions creates opportunities for sample depletion and may have impacted the performance of both experiments by lowering the abundance of total analytes in the samples.

It is important to note that the canine and HS-SPME-GC-MS experiments were not informed by the same sample sets. While all samples were collected by PVWDC and processed in the same manner, not all of the specific samples used to train or test detection dogs were used to inform the LDA model. Of the 291 samples used in canine training and 148 used to inform the computational models, 79 samples were used to inform both models. The increased number of samples used in training the canines may have beneficially impacted their ability to generalize the COVID-19 odor signature despite the presence of pet interferents.

Samples processed at FIU were subjected to UVC light exposure at 254 nm for 10-min prior to analysis. Hudson et al. (2009) noted that prolonged exposure to UVA/UVB light resulted in a distortion of human scent profiles, with some VOCs reducing in abundance and other compounds appearing as UV light exposure increased [53]. However, the Hudson study looked at direct, continuous exposure to the light source for 0, 1, 3, 5, or 7 weeks [53]. This study utilized a shorter exposure time of 10 min which has been previously reported to not adversely impact the VOC profiles being investigated [18].

### 4.3. Advantages and Disadvantages

In contrasting the use of canines and instrumental analysis to detect COVID-19 infection, the disadvantages of one approach appear to be compensated by the other in terms of mobility, speed of detection, development time, and performance rate. The HS-SPME-GC-MS method was developed using conventional benchtop GC-MS instrumentation with the HS-SPME procedure requiring heat baths and stabile working surfaces to prevent damage to the fibers while in use. The canines on the other hand are quite mobile once trained, they can easily be taken to new areas and be prepared to conduct their duties in under an hour where the movement of a GC-MS unit would require a day’s time in pull-down and set-up.

When viewing speed of detection, the canine method boasts a high-throughput rate with all five dogs being able to screen a sample in under ten minutes. In contrast, the HS-SPME method requires roughly 25 h for a sample to be analyzed and interpreted. In practice, samples are analyzed in batches to increase the efficiency of this method, the average per sample time equates to 3 h. While the canine method is much faster in its deployment, the average training time (per canine) was reported to be 31 weeks with an additional 5 weeks of testing. The HS-SPME-GC-MS method was developed in 16 weeks, owing 8 weeks to sample analysis and another 8 weeks to data interpretation and computational analysis. While the canine method may be faster to deploy, the HS-SPME-GC-MS method was quicker to develop.

In terms of ease of use, both methods require the participation of a trained expert with an analytical chemist being required to operate and interpret the results of a HS-SPME-GC-MS procedure and a trained canine handler being needed to conduct the canine association trials. Neither method is designed to be performed by a lay person.

Finally, when comparing performance at detecting COVID-19 infection, the final HS-SPME-GC-MS model demonstrated perfect performance with a LOOCV performance of 100% sensitivity and 100% specificity compared to the overall canine performance of 63% sensitivity and 90% specificity. The general performance of these two approaches is noted to be within a similar, competitive range as that of other previously published COVID-19 detection methods (Table 7). All in all, the disadvantages posed by one method were seen to be an excelling point in the paired approach, supporting the argument for a dual investigation of disease detection through canine and instrumental avenues.

### 4.4. Prospective Application of Research

This work demonstrated an approach to the dual investigation of canine and instrument-based detection of disease using volatile metabolomic expression. The instrumental model performed with high sensitivity and efficiency and was quick to train and implement. This approach has the potential to be translated into a miniaturized GC-MS approach or be used to inform the development of sensor device, applying the concept of an instrument-based detection system while improving upon mobility and speed of use issues. In fact, developments in chemiresistive sensors allow for their use with low-manufacturing costs, and ability to be incorporated into large sensor arrays for the detection of VOCs [61].

When viewing the canine approach, the performance of individual canines caused disparities in performance metrics. It is important to note that the canines participating in this study largely consisted of “pet dogs”. This is in contrast to such works as Grandjean et al. (2022), who utilized seven operational “working dogs” for their COVID-19 detection study and was able to achieve an average 97% sensitivity and 91% specificity [44]. This observation is indicative of the fact that canine performance is based on the aggregate performance of individual beings with varied abilities and that recruiting from a pre-filtered pool of professional detection dogs would likely improve the performance of this approach in its formal application as a diagnostic technology. A recent study by Kantele et al. (2022), which also implemented canines experienced in scent work, demonstrated great performance in the detection of a “wild-type” variant of COVID-19 with an overall 92% sensitivity and 91% specificity [58]. However, they also noted a decrease in performance when the canines were confronted with a previously unseen mutation of the SARS-CoV-2 virus. As such, it was implied that incorporating relevant COVID-19 variants in the training samples would allow for better overall performance.

It is believed that the ideal expansion and application of this work lies in the optimization of both approaches to reduce the identified deficiencies. The optimized methods would then be highly poised for joint application as a fast-screening approach implementing canine detection followed by a more accurate secondary analysis of suspected individuals via an instrument-based procedure. This joint approach of utilizing detection animals followed by benchtop instrumentation to improve the efficiency of delivering medical diagnoses has been demonstrated to be extremely effective as seen by APOPO’s efforts in tuberculosis detection using giant pouched rats, which reduces four days of analysis time to a 20-min screening and supplemental confirmatory testing [62]. Jointly accessing the speed of use of canines and high-performance capabilities of the instrumental approach is the ideal path forward for this dual assessment technique.

## 5. Conclusions

Together, the completed experiments demonstrate the capacity to simultaneously create and test instrumental and biological approaches for the detection of novel odor signatures. This ability was expressed through the discrimination of COVID-19 expression in body odor collected on T-shirts. Both experiments demonstrate the capacity of human body odor to be used as a sample source for determining COVID-19 status. The described methods were characterized by different assets: (a) a rapid, mobile approach implementing canines and (b) a reagent-free instrumental analysis method with high degrees of reproducibility. The two highest performing canines were able to distinguish the COVID-19 positive and negative individuals with an average performance of 88% sensitivity and 95% specificity; building upon this, the instrumental model (with pet interferents removed) reached an ideal performance of 100% sensitivity and 100% specificity. Ideally, in future work, both methods can be used in conjunction to be able to efficiently screen and detect diseases in a noninvasive manner. In this work, the issues that hindered one experiment were not seen to affect both. Moving forward, it is suggested that multiple relevant COVID-19 variants be included in the training samples for both approaches, as it was seen to affect diagnostic performance in a previous canine study. Further concurrent investigations of combined canine and instrumental detection schemes may shed light on how to overcome issues in one approach, informed by the other.

## Figures and Tables

**Figure 1 biosensors-12-01003-f001:**
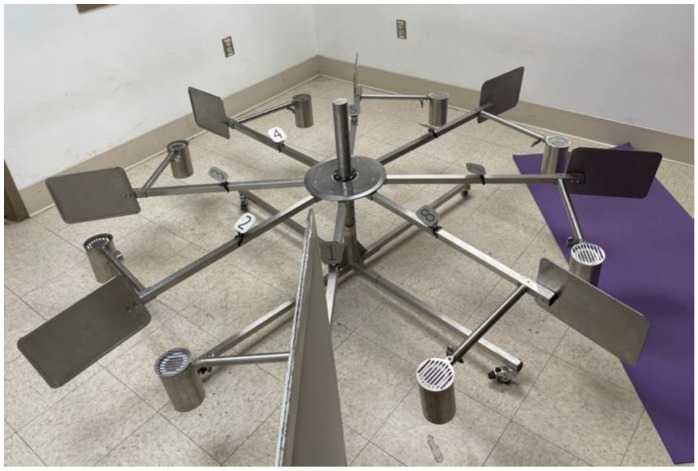
A photo of the 8-port scent wheel used in this study.

**Figure 2 biosensors-12-01003-f002:**
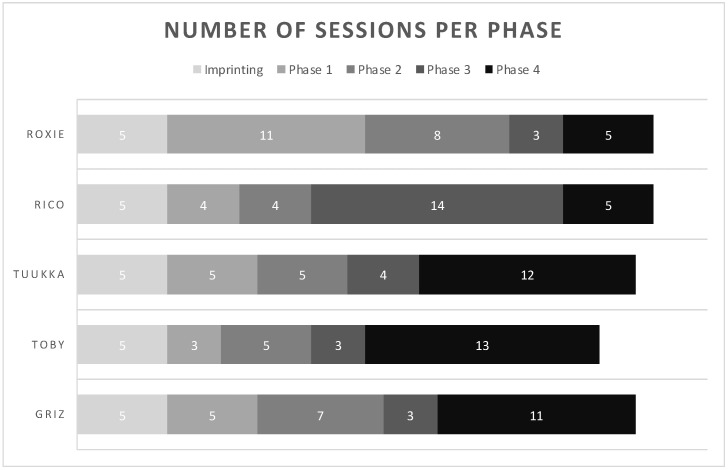
This graph shows the number of sessions each dog spent in each phase of training prior to the test phase. After imprinting, dogs had to spend a minimum of 3 sessions in each phase to meet the benchmark of 5/6 correct alerts on COVID-19 positive samples and 10/12 correct passes on COVID-19 negative samples.

**Figure 3 biosensors-12-01003-f003:**
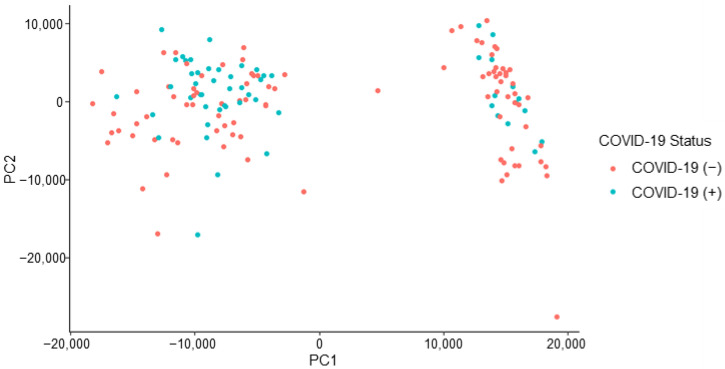
Initial COVID T-shirt PCA labelled by donor’s COVID-19 status.

**Figure 4 biosensors-12-01003-f004:**
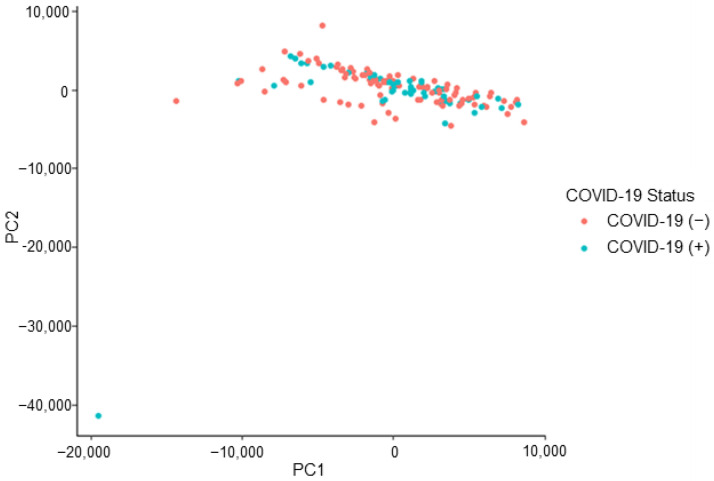
COVID T-shirt PCA: subgrouping removed, labelled by donor’s COVID-19 status.

**Figure 5 biosensors-12-01003-f005:**
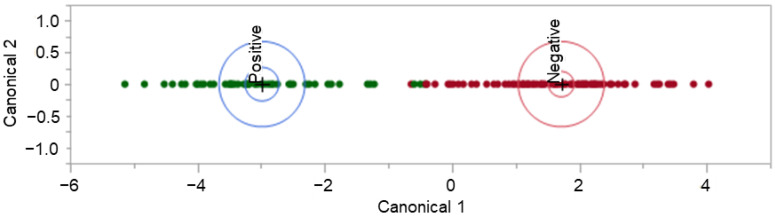
COVID T-shirt LDA model and model performance.

**Figure 6 biosensors-12-01003-f006:**
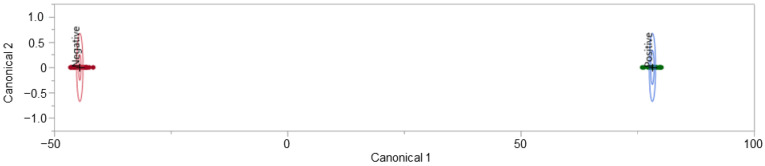
No pets in bed, COVID-19 T-shirt LDA model and model performance.

**Table 1 biosensors-12-01003-t001:** Donor demographic data for COVID-19 T-shirt samples used in dog training/testing.

Race/ Ethnicity	Identified Gender			COVID-19 Status/Presentation			Age Group			Shared Bed with				
	Female	Male	NB/GQ	Asym.	Sym.	Neg.	18–34	35–64	65+	No One	Another Person	Person and Pet(s)	Pet(s)	UKN
				*(+)*	*(+)*	*(-)*								
Asian or Asian American	12	1	-	-	3	10	11	2	-	5	5	2	1	-
Biracial	1	-	-	-	-	1	-	1	-	-	-	-	1	-
Black or African American	2	1	-	-	1	2	2	1	-	1	-	-	2	-
Hispanic or Latino	11	2	1	-	3	11	9	5	-	4	4	2	4	-
Native Hawaiian or Other Pacific Islander	1	-	-	-	-	1	-	1	-	-	1	-	-	-
White or Caucasian	206	45	1	9	69	174	120	118	14	82	70	36	62	2
Another race	5	-	-	-	2	3	4	1	-	2	-	1	2	-
Not reported	4	-	-	-	1	3	2	1	1	1	1	1	1	-
Total	239	49	2	9	79	202	147	128	15	95	79	42	72	2

NB/GQ = nonbinary/genderqueer.

**Table 2 biosensors-12-01003-t002:** Canine participant information.

Dog	Sex	Age (Years)	Breed	Previously Trained Odors
Griz	M	6	GSD	UDC, spotted lanternfly, Middle Eastern antiquities, live humans
Rico	M	6	GSD	UDC
Roxie	F	6	Labrador	UDC, narcotics, live humans
Toby	M	3	Small Munsterlander	UDC, spotted lanternfly, live humans
Tuukka	F	7	Husky/GSD mix	UDC

M = male, F = female, GSD = German shepherd dog, UDC = universal detector calibrant.

**Table 3 biosensors-12-01003-t003:** Training progression criteria.

Phase	Holes in Negative Lid	Sensitivity Required to Advance	Specificity Required to Advance	Minimum # Sessions Required to Advance	# COVID-19 (+) Samples/Session	# COVID-19 (-) Samples/Session
**Imprinting**	1	9/10 trials overall correct		1	1	1
**Phase 1**	1	80% (5/6 correct)	80% (10/12 correct)	3 in a row	2	4
**Phase 2**	4	80% (5/6 correct)	80% (10/12 correct)	3 in a row	2	4
**Phase 3**	16	80% (5/6 correct)	80% (10/12 correct)	3 in a row	2	4
**Phase 4**	Fully open	80% (5/6 correct)	80% (10/12 correct)	3 in a row	2	4
Sensitivity = TP/(TP + FN) Rate of alert to positive samples Specificity = TN/(FP + TN) Rate of no alert to negative/blank samples						

**Table 4 biosensors-12-01003-t004:** Donor demographic data for COVID-19 T-shirt samples informing computational models.

Race/ Ethnicity	Identified Gender			COVID-19 Status/Presentation			Age Group			Shared Bed With			
	Female	Male	NB/GQ	Asymp. (+)	Sympt. (+)	Neg. (-)	18–34	35–64	65+	No One	Another Person	Person & Pet(s)	Pet(s)
Asian or Asian American	5	-	-	-	1	4	4	1	-	2	3	-	-
Hispanic or Latino	2	2	1	-	3	2	3	2	-	1	1	1	2
White or Caucasian	100	25	0	3	37	85	58	57	10	36	36	19	34
Another Race	2	-	-		1	1	2	-	-	1	-	1	-
Not Reported	-	-	-		7 *^×^	4 ^×^	-	-	-	-	-	-	-
Total	109	27	1	3	49	96	67	60	10	40	40	21	36

NB/GQ = nonbinary/genderqueer * symptomology not reported. ^×^ demographic information not reported.

**Table 5 biosensors-12-01003-t005:** Dogs’ sensitivity and specificity at test.

Dog	Sensitivity	Specificity
Griz	11/12 * (91.7%)	51/56 * (91.1%)
Toby	11/13 (84.6%)	59/60 (98.3%)
Tuukka	9/13 (69.2%)	55/60 (91.6%)
Rico	5/13 (38.5%)	56/60 (93.3%)
Roxie	4/13 (30.8%)	46/60 (76.6%)

* Griz was accidentally clicked and rewarded after alerting on a positive sample during a test trial; because dogs were not being clicked and reinforced during testing, this trial was removed from his results before analysis.

**Table 6 biosensors-12-01003-t006:** Trial set individual positive results.

Trial	# of Dogs who Correctly Alerted	# Correctly Alerted, above Chance Dogs	Gender	Age	SymptomCount	Race/ Ethnicity	Bed Shared with Pet?
1	2/5 4/5	2/3 2/3	F M	45–54 35–44	5 5	White Hispanic	No Yes, 1 dog
2	4/5 3/5 3/5	3/3 1/3 2/3	F F M	25–34 35–44 55–64	3 6 5	White Hispanic White	No Yes, 1 cat No
3	3/5 3/5 4/5	2/3 3/3 2/3	F F F	25–34 25–34 55–64	4 0 7	Asian White African American	No No Yes, 1 or more dogs
4	2/5 3/5	2/3 3/3	M F	65+ 25–34	0 7	White Asian	No No
5	3/4 4/5 3/5	2/2 3/3 3/3	M F F	25–34 45–54 35–44	3 1 7	White White White	No No Yes, 1 or more cats

**Table 7 biosensors-12-01003-t007:** COVID-19 detection method comparisons.

Detection Method	Sample Medium	Average Specificity and/or Sensitivity	Reference
RT-PCR	Nasopharyngeal swab	98% sensitivity 100% specificity	[54]
	Saliva	69% sensitivity 100% specificity	[54]
Antigen test	Nasal swab	72.1% sensitivity 98.7% specificity	[55]
	Nasopharyngeal	65.7% sensitivity 100% specificity	[56]
Canine screening	Breath— face masks	83.1% sensitivity 88.6% specificity	[57]
	Face masks and clothes	86% sensitivity 92.9% specificity	[19]
	Skin swab	92% sensitivity 91% specificity	[58]
	Axillary sweat	97% sensitivity 91% specificity	[44]
	Axillary sweat	89.6% sensitivity 83.9%specificity	[57]
	Body odor—T-shirts (including axillary sweat)	63% sensitivity 90% specificity	Current study
HS-SPME-GC-MS	Body odor—T-shirts (including axillary sweat)	100% sensitivity 100% specificity	Current study
SPME-GC-MS	Blood serum	94% sensitivity 83% specificity	[59]
Quartz microbalance	Blood serum	94% sensitivity 80% specificity	[59]
Colorimetric paper sensor	Breath	78.3% sensitivity 83.6% specificity	[60]

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
