# Peer review of "The Use of Biological Sensors and Instrumental Analysis to Discriminate COVID-19 Odor Signatures"

_biosensors, 2022, doi:10.3390/bios12111003_

Round 1
Reviewer 1 Report
It is advised to use the terms positive and negative instead of + and - (COVID +) throughout the manuscript.
The Introduction must be revised by adding a few more recent findings and references on the proposed study topic.
Please highlight the major drawbacks and shortcomings of the proposed approach.
Articles suggested to cite for better readability:
- https://doi.org/10.1080/23744235.2022.2033311
-
- https://doi.org/10.1186/s12879-021-06523-8
- - https://doi.org/10.3390/diagnostics12020430
The Conclusions must be revised by adding strong points and take-home-message from the results obtained.
Author Response
Reviewer #1
Thank you for the time you have taken to review our article and for your thoughtful input. We have done our best to address the issues or need for additional clarity you have pointed out. Many of our additions are made in the direct text (see referenced sections). We believe your recommendations have improved the quality of this manuscript and are grateful for your help in the revision process.
- It is advised to use the terms positive and negative instead of + and - (COVID +) throughout the manuscript.
The terms COVID-19 (+/ -) have been switched to COVID-19 positive and COVID-19 negative.
- The Introduction must be revised by adding a few more recent findings and references on the proposed study topic.
The introduction has been updated with more recent information and research findings.
- Please highlight the major drawbacks and shortcomings of the proposed approach. Articles suggested to cite for better readability: - https://doi.org/10.1080/23744235.2022.2033311 - https://doi.org/10.1186/s12879-021-06523-8 - https://doi.org/10.3390/diagnostics12020430
Please see section 4.3.
- The Conclusions must be revised by adding strong points and take-home-message from the results obtained.
Please see the updated conclusion. A future impact statement has been added as well as the overall performance summary of the approach to reiterate its potential impact in a future application.

Reviewer 2 Report
The manuscript ID biosensors-1964922 mainly presents a study about the performance of dogs and the headspace analysis of COVID-19 samples by noninvasive headspace-solid phase microextraction-gas chromatography-mass spectrometry. A list of comments for the authors is below:
1. A graphical abstract illustrating the methodology and the main findings would be welcome.
2. For section 2.1.2, the season of the year for the experiments could be an influence in the main results? Please argue.
3. The discussion section must highlight the advantages and disadvantages of this proposed methodology for the use of biological sensors and instrumental analysis to discriminate COVID-19 odor signatures.
4. Please describe some perspectives in the use of this proposed methodology to be a base for future research. The authors are invited to see for instance the use of machine learning as described by: https://doi.org/10.3390/bios12090710
5. If possible, it would be helpful to comment about the evolution of these observations taking into account the well-known mutation of the SARS-CoV-2.
6. A confrontation of the main results with updated publications in the same topic could be helpful for readers. You can see for instance: https://doi.org/10.3390/chemosensors10060199
7. Do the participants were vaccinated? Do vaccinations represent a contribution for the main observations?
8. Please deeper explain the dispersed data depicted in figure 4.
9. Some references should be updated. Please see reference 1.
10. It is suggested to split some of the collective citations in order better present the topic, and to see the individual justification and importance of each reference selected to be part of this manuscript.
Author Response
Thank you for the time you have taken to review our article and for your thoughtful input. We have done our best to address the issues or need for additional clarity you have pointed out. Many of our additions are made in the direct text (see referenced sections). We especially appreciate the suggestion of adding a graphical abstract, as we had not previously considered including one. We believe your recommendations have improved the quality of this manuscript and are grateful for your help in the revision process.
Please see the attached word document for specific responses and the new graphical abstract.

Round 2
Reviewer 2 Report
The manuscript has been improved. However, please see below the following issue:
*The discussion section and perspectives must highlight the advantages and disadvantages of this proposed methodology. The use of biological sensors and instrumental analysis to discriminate COVID-19 odor signatures with a base on confrontation taking into account updated publications can be considered. The comments provided in the edition are interesting but unspecific; they should be supported. A table with numerical data confronted with other techniques would enhance the presentation of the work.
Author Response
Thank you for your continued interest in our findings and efforts to improve upon our manuscript.
We have incorporated an additional table (Table 7) to more concisely exhibit a direct performance comparison between the canine and instrumental analysis approaches employed in this study and the reported sensitivity/ specificity rates of other peer-reviewed published findings. We believe this table acts as a direct comparison point for the efficacy of the studied approaches compared to other published findings in Covid-19 detection. We have made sure to include a mixture of detection methods for a breadth of reference points.
Round 3
Reviewer 2 Report
I agree with the potential impact of reviewed version of the manuscript. Then I can recommend this work for publication in present form.